Ecosystem antifragility: beyond integrity and resilience

http://orcid.org/0000-0001-5306-7397 Equihua Miguel 1
http://orcid.org/0000-0002-8934-375X Espinosa Aldama Mariana 2
http://orcid.org/0000-0003-0193-3067 Gershenson Carlos 3 4 5
http://orcid.org/0000-0002-2926-7791 López-Corona Oliver 1 4 6 olopez@conacyt.mx
Munguía Mariana 7
http://orcid.org/0000-0002-4528-3548 Pérez-Maqueo Octavio 1
Ramírez-Carrillo Elvia 8 elviarc@otrasenda.org
1 Red Ambiente y Sustentabilidad, Instituto de Ecología A.C. , Xalapa, Veracruz , México
2 Doctorado en Ciencias Sociales y Humanidades, UAM-Cuajimalpa. , CDMX , México
3 IIMAS, Universidad Nacional Autónoma de México , CDMX , México
4 Centro de Ciencias de la Complejidad (C3), Universidad Nacional Autónoma de México , CDMX , México
5 ITMO University , St. Petersburg , Russia
6 Cátedras CONACyT, Comisión Nacional para el Conocimiento y Uso de la Biodiversidad (CONABIO) , CDMX , México
7 Comisión Nacional para el Conocimiento y Uso de la Biodiversidad (CONABIO) , CDMX , México
8 Facultad de Psicología, Universidad Nacional Autónoma de México , CDMX , México
Hoover Kara
Electronic publication date: 2020 Feb 11
Publication date: 2020
Volume: 8
Electronic Location ID: e8533
Received 2019 Jun 20; Accepted 2020 Jan 7
Copyright: © 2020 Equihua et al.
Copyright year: 2020
Copyright holder: Equihua Zamora et al.
License: This is an open access article distributed under the terms of the Creative Commons Attribution License, which permits unrestricted use, distribution, reproduction and adaptation in any medium and for any purpose provided that it is properly attributed. For attribution, the original author(s), title, publication source (PeerJ) and either DOI or URL of the article must be cited.
License URL: https://creativecommons.org/licenses/by/4.0/

Keywords: Antifragility, Ecosystem integrity, Resilience, Complexity

Funding: CONACyT Fund M0037-2018-07 296842 Cátedras CONACyT Fellowship Program 30 Sistema Nacional de Investigadores SNI 62929 This work was supported by CONACyT fund M0037-2018-07, Number 296842, Cátedras CONACyT fellowship program (Project Number 30), and Sistema Nacional de Investigadores SNI, Numbers 62929. The funders had no role in study design, data collection and analysis, decision to publish, or preparation of the manuscript.

==============================
We review the concept of ecosystem resilience in its relation to ecosystem integrity from an information theory approach. We summarize the literature on the subject identifying three main narratives: ecosystem properties that enable them to be more resilient; ecosystem response to perturbations; and complexity. We also include original ideas with theoretical and quantitative developments with application examples. The main contribution is a new way to rethink resilience, that is mathematically formal and easy to evaluate heuristically in real-world applications: ecosystem antifragility. An ecosystem is antifragile if it benefits from environmental variability. Antifragility therefore goes beyond robustness or resilience because while resilient/robust systems are merely perturbation-resistant, antifragile structures not only withstand stress but also benefit from it.

Introduction

Sustainable development needs to preserve the structure and functioning of natural ecosystems, that is, their integrity, as a sine qua non condition. In previous work (Equihua et al., 2014) an operational framework has been developed (see Fig. 1) to quantify ecosystem integrity as well as viable standards useful for managing the way ecosystem interventions, promoting development along sustainable avenues.

Figure 1 Ecosystem integrity three-tier model.

Ecological integrity is understood to be an underlying attribute in the constitution of ecosystems that produce specific manifestations in their structural characteristics, development processes and acquired composition. In short, ecosystem integrity arises from processes of self-organization derived from thermodynamic mechanisms that operate through the locally existing biota, as well as the energy and materials at their disposition, until attaining “optimal” operational points which are not fixed, but rather vary according to variations in the physical conditions or changes produced in the biota or the environment. we show the three-tier model of ecosystem integrity (3TEI), the inner tier is hidden to the observer, but its status can be inferred by the information available at the instrumental or observational tier where measurements on structure (including composition or other biodiversity features) and function are obtained, of course considering the context where the ecosystem is developing. Arrow tips indicate the direction of assumed mechanistic influence, although information can go either way.

Humans are starting to be recognized as an overwhelming forcing factor modulating biosphere dynamics. In this view, the Earth system can be interpreted as entering in a geological era that can be called the Anthropocene (Steffen, Crutzen & McNeill, 2007), the Technocene (López-Corona, Ramrez-Carrillo & Magallanes-Guijón, 2019), or even Capitalocene (Haraway, 2015), depending on the conceptual stance adopted. Because of this driving influence of human decisions, there has been long interest in understanding and measure the way ecosystems recover (or not) from human perturbations. To operationalize sustainability, we require working definitions of this ecosystem ability and metrics to asses it, in addition to ecosystem integrity (as a component of a likely state variable accounting for the amount of natural capital assets: condition × extension). The recovery property has already been encapsulated into the ecosystem resilience concept. Here, we propose that information theory is a suitable framework to encompass both ecosystem integrity and resilience as we will discuss in next subsection.

According to Equihua et al. (2014), numerous studies have aimed to find a suitable and inclusive definition of ecological integrity; however, no general consensus has been achieved to date. In their work, Equihua and co-workers embrace a complex systems approach in which ecosystems are considered as self-organized entities constrained in their structure (including biological composition or biodiversity), and function by thermodynamic dissipative system properties (Kay, 1991; Regier, 1995; Manuel-Navarrete, Kay & Dolderman, 2007; Michaelian, 2005) and evolutionary processes (Levin, 2005).

The concept of ecological resilience was first introduced by Holling (1973) to portray the persistence of natural structures in the presence of environmental stressors due to natural or anthropogenic triggers. In his seminal work, Holling follows a system dynamics analysis. There, resilience is defined as: “the capacity of a system to absorb disturbance and reorganize while undergoing a change so as to still retain essentially the same function, structure, identity, and feedbacks” (Walker et al., 2004). Nevertheless, resilience or stability, which is commonly used as synonym, have at least 163 different definitions (grouped in at least 70 concepts of stability/resilience) (Grimm & Calabrese, 2011). For example, Saint-Béat and co-workers (Saint-Béat et al., 2015) summarize resilience and others related concepts often used interchangeably as follows: Resilience is the rate at which a system returns after a disturbance to the equilibrium state (DeAngelis, 1980; Pimm & Pimm, 1991). Long return time is equivalent to low resilience. A community’s resilience relies on the least resilient species (the slowest to return to equilibrium). This definition of resilience corresponds to the “engineering resilience” defined by Holling (1996), and assumes that there is only one balance or a stable state (Gunderson, 2000).

Persistence is the time for a variable to remain in the same state before changing to a different one (Pimm & Pimm, 1991). Persistence is a measure of a system’s capacity to preserve itself over time (Loreau et al., 2002).

Resistance is described as the capacity of an ecosystem in the presence of external disturbance to preserve its initial state. (Harrison, 1979). Only small changes (in amount and intensity) within an ecosystem correspond to high resistance. This concept is similar to the “ecological resilience” defined by Holling (1996) and suggests that various stable states exist.

Robustness relates to the durability of the stability of the environment. Robustness is then a measure of the amount of disturbance an ecosystem can endure before it changes to a different state. (Loreau et al., 2002). The more robust the food web is, the more stable it is.

Information theory as unifying framework

Ecosystem integrity arises from processes of self-organization derived from thermodynamic mechanisms that operate through the locally existing biota, as well as the energy and materials at their disposal, until attaining optimal operational points which are not fixed, but rather vary according to variations in the physical conditions or changes produced in the biota (Equihua et al., 2014).

In this work we want to highlight the interaction between ecosystem integrity and its response to perturbations. The standard way for understanding how ecosystems respond to perturbations is ecological resilience or the magnitude of perturbations that ecosystems can withstand and return to the target state, as opposed to shifting into an alternative stable state.

Although both concepts are related they are not the same. It is clear that external perturbations affect the state of integrity of the ecosystems and similarly the integrity state reflects on the ecosystem capacity for coping with perturbations. However, for example, we may have an ecosystem that has been undergoing a degradation processes such as defaunation but still manifests a high integrity in terms of structure and composition but has only one species in a functional group. In this case, as the function is still represented, the ecosystem maintains its integrity but it would be very fragile under perturbations that affect that key species. Thus, integrity and resilience are not the same. The first is a static picture of the ecosystem and the second is much more related with ecosystem dynamics.

In his seminal article form 2003, Ulanowicz pointed out that, unlike classical mechanics that are fully described by Newton’s Laws or Electromagnetic theory from Maxwell’s equations, more than a century after it first appeared on the scene, ecology today still appears to be too diverse and conflicted to be able to coalesce around any one coherent theory (Ulanowicz, 2003).

Nevertheless, it has been proposed that indicators based on information theory can bridge the natural and human system elements, and make sense of the disparate state variables of the system (Cabezas et al., 2005). Moreover, information theory has played a main part in our knowledge of biological systems and it is not new to see ecosystems as information systems themselves (Nielsen, 2000; Straskraba, 1995). For instance, several authors such as Brooks & Wiley (1984), Wicken (1987) and Michaelian (2012) have pointed out the importance of entropy, thermodynamics and information in evolutionary processes.

In the same way, as explained by Fath, Cabezas & Pawlowski (2003), as ecosystems develop, they move further from equilibrium. One hypothesis posed, is that this results in organisms with greater genetic information capacity (Jørgensen & Nielsen, 1998). Therefore, this use of information combines the internal make-up of individual organisms with the overall ecosystem’s structural organization. Even more, all living systems need to acquire information from their environment and generate mechanism to respond and adapt to changes on it. For this reason, biological systems are constantly modeling their environments in internal representations based on data available through their sensors. In this context better hardware (sensors) will translate into advantages, but it is the successful construction of these representations, which extract, summarize and integrate relevant information, the one that provides a crucial competitive advantage, which can eventually make the difference between survival and extinction. In fact, this feature might be so important that Hidalgo et al. (2014) have proposed ecosystems tends to be in a critical state in which this computational capacities are maximal.

Considering the above, we think that both ecosystem integrity and how ecosystems respond to perturbations are two separate dimensions necessary to assess the condition of ecosystems and that information theory can provide a general unifying framework for understanding both.

With this in mind, we realized a review with the key words “ecosystem integrity” AND “ecosystem resilience” AND “information theory” because the first two are different dimensions of ecosystems dynamics that interact with each other in a very coupled but not trivial way, and information theory is an adequate general framework for understanding each separately and their interactions. This review exercise is summarized in the Supplemental Materials and we will focus here in discussing the main narratives found and the most important, in the proposal of a new concept that could replace ecosystem resilience because it is more general and better supported by an Information theoretical framework.

Literature narratives

We consider that in the context of the relation of resilience with ecological integrity under the lens of information theory, narratives has gone from trying to identify (with information theory tools) ecosystem features that allow them to be more resilient and hence maintain their integrity, to a more technical approach using times series or network analysis in addition to new mathematical concepts (see Fig. 2). The importance of interactions and a complex system approach is highlighted. Finally, the field completes a circle getting again back into ecology and refining resilience feature and properties of ecosystems.

Figure 2 Summary of concepts and narratives in selected papers.

Ecosystem properties that enable them to be more resilient

As we have discuss in the Introduction, it is intuitive that inner properties of the ecosystems may determine both its integrity and resilience. In this subsection we summarize how this idea is represented in the main literature narratives trying to make clear the connections between both concepts and information theory.

Lets first consider the revised article by Aronson & Le Floc’h (1996), where the authors refer to a previous work (Aronson et al., 1993) in which they modified the Noble & Slatyer (1980) concepts, who described several categories of essential properties that determine a species reaction to recurrent disturbances.

In the Aronson article, the authors defined vital ecosystem attributes (VEAs) summarized in Table 1 as an attempt to capture those features or attributes that are related with and that represents ecosystem structure and function, the same type of attributes used in the ecosystem integrity Model (3TEI) (Equihua et al., 2014). In terms of ecosystem resilience, one may interpret that these VEAs fall into optimal values when the ecosystem is more resilient. An interesting point is that unlike the 3TEI model, VEAs requires intense (often cost and not scalable) fieldwork that most likely make VEAs not such a good option for a national assessment of ecosystem condition trends.

Table 1 Vital ecosystem attributes according to Aronson et al. (1993).

Structure	Function	
Perennial species richness	Biomas productivity	
Annual species richness	Soil organic matter	
Total plant cover	Maximum available water reserves	
Aboveground phytomass	Coefficient of rain off efficacy	
Beta diversity	Rain use efficacy	
Life form spectrum	Length of water availability period	
Keystone species	Nitrogen use efficacy	
Microbial biomass	Microsymbiont effectiveness	
Soil biota diversity	Cyclic indexes	

From the idea of VEAs, Aronson & Le Floc’h (1996) identified also 16 quantifiable key characteristics for use on a landscape scale, to use the fresh vital landscape attributes (VLAs) to evaluate the outcomes of ecological restoration or rehabilitation conducted from the view of the landscape. Most interesting is the chance that as VEAs relate to the integrity of the ecosystem, VLAs (see Table 2) could be related to a landscape integrity (3TEI) model, we are already working on.

Table 2 Vital landscape attributes as proposed by Aronson & Le Floc’h (1996).

Vital landscape attributes (VLAs)	
Type, number and range of landform	
The number of ecosystems	
Type, number and range of land units	
Diversity, length and intensity of former human uses	
Diversity of present human uses	
Number and proportions of land use types	
Number and variety of ecotunes-zones	
Number and types of corridors	
Diversity of selected critical groups of organisms (functional groups)	
Range and modalities of organisms regularly crossing ecotunes	
Cycling indexes of the flow and exchanges of water, nutrients, and energy within and among ecosystems	
Pattern and tempo water and nutrient movement	
Level of anthropogenic transformation of landscape	
Spread of disturbance	
Number and importance of biological invasions	
Nature and intensity of the different sources of degradation, whether legal or illegal	

In a different line of thought, the reviewed article by Gustavson, Lonergan & Ruitenbeek (2002) and co-workers develops a general index that may serve as a proxy of ecosystem resilience from an information theory perspective. The authors report that attempts have been made to describe and evaluate resilience, but an overall predictive or theoretical connection between resilience characteristics and ecosystem dynamics has yet to be advanced. Similarly, much has been discussed about possible interactions between stability and structure. Generally speaking, predictable interactions between resilience characteristics and how ecosystems work are not intuitively evident and may not exist.

To this end, they turn to Ulanowicz’s ascendancy theory (Ulanowicz, 1986) which is a measure of the magnitude of the information flow through an ecosystem’s network framework. One constraint for its use is that it requires a comparatively full description of the nature and magnitude of all species interactions.

Ascendancy is defined by the average mutual information (Ulanowicz, 1986) between component a to b, (1) As=K∑i⁡∑j⁡p(ai,bj)log[p⟨bj|ai⟩p⟨bj⟩]

where, p⟨bj|ai⟩, the probability of bj given that ai has occurred; and p(bj), the probability that bj will occur.

The upper limit of ascendancy is the capacity for growth, and the distinction between ability and ascendancy is called the overhead system, which represents a multiplicity of paths and can therefore eventually be linked with the complexity of the ecosystem (Zorach & Ulanowicz, 2003).

A complex structure is key for ecosystem integrity and resilience because the diversity (i.e., measured by Shannon information) of process, associated with complexity as we will defined later, plays a crucial role on system survival (Ulanowicz et al., 2009). In particular, to enhance ecosystem’s long-term integrity and resilience, a particular densely connected network structure is advantageous. Such a scheme is sufficiently effective and sufficiently varied.

Ultimately, ascendancy captures in a single index the capacity of an ecosystem to prevail against disruption by virtue of its combined organisation and size, it was suggested that, in order to attain sustainability, ecosystems should be able to return to the ascendancy levels before perturbation (Reynolds, 2002).

At this point, it is convenient to clarify some aspects. Ecosystems don’t have a unique attractor which is a subset of states that the ecosystem tends to evolve to for a wide variety of starting conditions. A useful metaphor could be an egg carton structure in which a ball is placed into one of the carton basins (attractors). One may perturb the ball but it returns to the same basin until a point in which it go out and fall in a new basin. As we have been arguing, the capacity of the system to return to a specific attractor, its resilience and is related with ecosystem integrity. It would be expected that higher levels of integrity are related with a higher resilience too. Nevertheless, on the one hand this relation is most probably non linear in the sense that the ecosystems may experiment important integrity loss before resilience reflects it. On the other hand, the ecosystem may have a relatively high level of integrity but for example, there could be a unique species in a specific functional group, then if the perturbation eliminates this species the system may lose its resilience all at once. A weaker concept of resilience is associated with the so called engineering resilience that is measured with the time a system requires to return to a particular attractor. In this work when talk about resilience we mean ecological resilience not merely engineering resilience.

In the third article revised, the main purpose of Saint-Béat et al. (2015) is to know how distinct ecosystems react to global change in terms of composition, dynamics and eventually, how persistence, strength, or resilience of the ecosystem can be assessed.

The authors show that ecological network assessment (ENA) offers an effective approach for describing local stability, combining both quantitative and qualitative elements. They warn, however, that describing real cases combining local and global stability remains an incomplete task.

They focus then on three resilience-related results that emerge from their ENA: (a) the role of species diversity in the structure and functioning of the ecosystem; (b) the number of trophic links and strength of interactions; (c) the stability of the ecosystem in terms of cycling capacity, omnivory spread and ascendancy.

High biodiversity (without considering non-native species) was proposed to contribute to minimize the threat of major ecosystem modifications in reaction to environmental disturbances (McNaughton, 1977). Experiments on species invasion in grassland parcels indicate that local biodiversity decreases the settlement and success of a variety of invaders (Kennedy et al., 2002). Similarly, studies on manipulation of grassland diversity indicate that elevated diversity improves inter-specific competition and thus decreases the danger of invasion (Naeem et al., 2000; Hector et al., 2001). In the same way, it has been showed that species diversity also contributes to enhance resilience in coastal ecosystem (Worm et al., 2006). Similarly, the preservation of species diversity seems crucial to achieving the maintenance of Ecosystem Integrity and the services they provide that are critical to human societies (Chapin Iii et al., 2000).

However, the authors warns that the consideration of a single variable such as diversity can not be sufficiently to evaluate the stability of ecosystems due to its complexity. Therefore, they suggest that building holistic indexes in the ENA framework is a better approach for a thorough knowledge of the food web structure and its role in the functioning of the ecosystem.

If one would be able to construct a sufficiently detailed trophic network (something very difficult to do in general), one could use standard network analysis tools to understand, for example, the topology of the network (i.e., connectance). In that sense, the authors summarize evidence from the literature to show that an increase in links dissipates the impact of variability in species distribution, increasing stability (MacArthur, 1955). Higher connectivity thus improves also ecosystem resilience (DeAngelis, 1980). Then, connectivity seems to be a useful measure of the robustness of food webs and indirectly of ecosystem stability.

In addition to connectivity, they show the importance of interaction strength diversity. Following ideas of Ulanowicz (1983), they claim that ecosystem stability requires a balanced presence of weak and strong interactions. Moreover, the food web would be stable if and only if main predator-prey interactions are combined with weak interactions in the context of high diversity. Thus, due to their ability to fluctuate and adapt within ecosystems, weak interacting species function as a stabilizing force in food webs and consequently the ecosystem.

To gain a deeper understanding of stability, Saint-Béat et al. (2015) examine the effect of cycling, the presence of omnivorous and ascending.

For instance, the presence of omnivory gives the ecosystem trophic flexibility, a clear beneficial feature that reflects Ecosystem Integrity and Resilience. The researchers claim that omnivory provides the ecosystem a superior buffer to deal with environmental disturbances because they enable faster ecosystem reaction by rapidly moving trophic routes following disturbance. For example, if a disturbance impacts low trophic levels, omnivorous species that are directly linked to it, would respond rapidly. In comparison, a particular predator must wait until the disturbance reaches its own level; therefore, the response time will be longer.

As in the reviewed article by Gustavson, Lonergan & Ruitenbeek (2002), Saint-Béat et al. (2015) show how ascendency could be used as a key indicator to evaluate ecosystems functioning. The authors indicate that to understand the function of “ascendancy” two kinds of stability must be differentiated. A system with elevated inner stability is a system with adequate inner constraints to enable a strongly organized structure, corresponding to a high ascendency (high mutual information). Typically under this condition, ecosystems are some how protected against internal perturbations but leave them vulnerable to external ones. On the other hand, since low ascendancy is linked to redundancy, ecosystems become more resilient to external disturbances. Interestingly enough, too high level of ascendancy is recognized as a characteristic of stress and may indicate a decrease ecosystem resilience.

Summarizing, in the dynamic response of ecosystems under the criticality framework (Ramírez-Carrillo et al., 2018), a healthy ecosystem is found where a balance between robustness and adaptation is reached. In the case of network topology, the ecosystems need to develop a good balance between strong and weak interactions in order to be stable. In the case of the ascendancy narrative, a stable ecosystem should develop a good balance between ascendancy and overhead, which seems to give resistance and resilience to ecosystems.

This leads us to think that all these three kinds of balance could be particular cases of a more general evolutionary strategy of living systems: the antifragility, which will be discuss in a following section.

Ecosystems response to perturbations

The main interest in the selected article form Cabezas et al. (2005) is not ecosystem integrity nor resilience per se but sustainability. Nevertheless, they emphasize that the structures and operation of the human component (in terms of culture, economy, law, etc.) must be such that they enhance the persistence of the natural component’s structures and operation (in terms of ecosystem trophic connections, biodiversity, biogeochemical cycles, etc.), and vice versa. It is in the idea that “persistence” and “operation of the natural component” that the connection is made. From their perspective a sine qua non condition to achieve sustainability (Ramírez-Carrillo et al., 2018) is ecosystem stability which they conceptualize using Fisher information (Cabezas & Fath, 2002a), view that is further developed in several papers (Eason & Cabezas, 2012; Karunanithi et al., 2008; Mayer et al., 2007, Mayer, Pawlowski & Cabezas, 2006; Zellner et al., 2008; Ahmad et al., 2016; Karunanithi et al., 2011; Gonzalez-Mejia et al., 2012) using an informational theory approach based mainly on Fisher information (Mayer, Pawlowski & Cabezas, 2006).

To understand stability or resilience, lets follow Binder (2000) and Mayer et al. (2007) and consider the central problem of estimating the actual value θ for a state variable. The estimation comes from an inference process from imperfect observation y = θ + x in the presence of some random noise x. This kind of measurement-inference process of θ whose result is an estimator θ^ is always the result of an imperfect observation θ^(y) for which the mean-square error obeys the Cramer–Rao inequality (2) e2I≥1

where I is the Fisher information of the system, calculated as (3) I=∫dyP0(y|θ)[dP0(y|θ)dθ]2

in which P0(y|θ) is the probability density function of measuring a particular value of y given the true value θ of the state variable in question.

Then, since the error decreases as information increases, Fisher information may be understood as the quality of the estimation θ from the measurement-inference process. Then, if the system is characterized by a phase space with m state variables xi that define the phase vector s = (x1,…, xi,…, xm) associated with a smart measurement y, then we can prove that (4) I(s)=1T∫T0s″2s′4dt

where T is the time period required for one cycle of the system; s′(t) is the tangential speed and s″(t) is scalar acceleration tangential to the system path in phase space. Both are calculated in terms of the state variables xi as (5) s′(t)=∑im(dxidt)2

(6) s″(t)=1s′(t)∑im(dxidtd2xidt2)

A simple and robust approach to calculating tangential velocity and acceleration uses the three-point difference scheme (7) dxidt=αxi(t+Δta)−(α2−1)xi(t)−xi(t−αΔta)α(α+1)Δta

(8) d2xidt=αxi(t+Δta)−(α+1)xi(t)−xi(t−αΔta)α(α+1)Δta2/2

where xi(t) is a central data point, xi(t − Δta) is the next point following the center xi and xi(t − Δtp) is the previous point to it. For evenly-spaced points Δta = Δtp and α = Δtp/Δta is the ratio of the previous and following time space.

The thesis suggested by Frieden (2007) and Cabezas & Fath (2002b) is that a shift in Fisher information may signal a change of regime in a dynamic system in which: Fisher information is a function of measurement variability. Low variability results in high Fisher information and low Fisher information results in high variability.

Systems in stable regime tends to exhibit constant Fisher information. Then, organization losses points to greater variability and a decrease of Fisher information.

Self-organizing systems reduce their variability and gain Fisher information.

“If resilience is defined by the intensity, frequency, and duration of a perturbation that a system can withstand before fundamentally changing in function and structure, then we would hypothesize that Fisher information would return to the same value or higher in more resilient systems” (Cabezas et al., 2005).

We found this last point of much interest because not only it provides a formal informational definition of ecosystem resilience, but it also provides a specific way to measure it via Fisher information. In order to test this idea, we analyze NDVI data for “US-Me1: Metolius—Eyerly burn” Ameriflux site (Guy, Kosugi & Sulzman, 2007) site in Oregon for which is documented as an intermediately aged ponderosa pine forest that was severely burned in the 2002 Eyerly wildfire. The AmeriFlux network of approximately 100 research stations is the main research group and information supplier for big terrestrial carbon cycling syntheses in the Americas and has established a database for micrometeorological, meteorological and biological information.

Data of NDVI was downloaded using the application for Extracting and Exploring Analysis Ready Samples (AppEEARS) that enables users to subset geospatial data-set using spatial, temporal, and band/layer parameters (https://lpdaacsvc.cr.usgs.gov/appeears/). In particular, we used MOD13A3.006 1 km2 resolution monthly data of NDVI from 01-01-1990 to 01-01-2018.

Focusing into the 2002 wildfire, we show in Fig. 3 that as expected, with the wildfire disturbance the system experience both great changes in NDVI and its corresponding Fisher information. We found that Fisher information returns to previous values after 18 months approximately but not the NDVI values.

Figure 3 In red the normalized NDVI time series for the 1 km2 pixel corresponding to the coordinates of the US-Me1 site of Ameriflux with a monthly sampling.

In blue, the corresponding values of Fisher’s information using the Cabezas and collaborators algorithm (https://github.com/csunlab/fisher-information).

On the one hand, recovering Fisher information could be related to the way Filotas and co-workers (Filotas et al., 2014) understands ecological resilience as “the amount of change that an ecosystem can absorb before it loses its ability to maintain its original function and structure.” After a disturbance, the authors claim that a resilient system has the ability to recover its initial structure, features, and feedback; in other words, its integrity.

On the other hand, it seems then that the criterion of Fisher information is necessary but not sufficient to ensure ecosystem resilience. For example one should expect that after a disturbance, essential variables as the ones proposed by Schmeller et al. (2018), which could be seen as a modern version of Aronson’s vital ecological attributes (Aronson & Le Floc’h, 1996), return to previous values. In a recent work Dutrieux (2016) combine into one index signals from TM and ETM+ B4 band, corresponding to near infra red (NIR) with wavelength of 770 − 900 nm which provide information about canopy biomass; and B5 band corresponding to short wave infra red (SWIR1) with wavelength of 1550 − 1750 nm which provide information about canopy moisture content: (9) NDMI=NIR−SWIR1NIR+SWIR1

Low NDMI values for bare soils and thin forest canopies are anticipated, while greater values correspond to thicker, completely developed forest canopies. (Wilson & Sader, 2002).

The author created the following harmonic model to compare values before and after a disturbance: (10) yt=αi+∑13γisin(2πjtf+δj)+εt

where the dependent variable y at a given time t is expressed as the sum of an intercept αi, a sum of different frequency harmonic components representing seasonality and an error εt. In the model j corresponds to the harmonic order, one being the annual cycle, γj and δj correspond respectively to the amplitude and phase of the harmonic order j, and f is the known frequency of the time-series (i.e., number of observations per year).

New values are then estimated for each spectral band and each time series observation using the corresponding matched model, enabling Euclidean distance to be calculated with the following formula: (11) Dt=∑i=1k(y^it−yit)2

The author then applies it to spectral recovery time for a set of 3,596 Landsat time-series sampled from regrowing forests across the Amazon basin, thus producing estimates of recovery time in spectral properties, which he calls spectral resilience. On average, he found that spectral resilience takes about 7.8 years, with a large variability (SD = 5.3 years) for disturbed forests to recover their spectral properties.

Now we have a new problem, how to determine the thresholds for (a) distance between initial and final values for both state (essential/vital) variables and their Fisher information; (b) the time scale these recovering should occur. In principle we believe (a) could be determined from ecological integrity measurements, but it is currently an open research question we are not addressing here. For (b) let us remember that this returning time is only important in terms of current ecosystem state but not as a measure of globally stability—that is, when another attractors exists.

In another line of thoughts, Sidle and co-workers (Sidle et al., 2013) focus on ascertaining under what circumstances ecosystems exhibit resilience, tipping points or episodic resetting. They point out that while ecosystem resilience originated from ecological perspective, latest debates incorporate a geophysical context where it is acknowledged that dynamic system properties may not return to their former state after disturbances (Dakos et al., 2018; Gough et al., 2017; Langdon et al., 2016; Ravindran, 2016; Moore, 2018; Steffen et al., 2018). This phenomena known as Tipping points, generally arise when chronic (typically anthropogenic but sometimes natural) changes push ecosystems to thresholds that cause process and function collapse even in a permanent way. Resetting ecosystems happens when episodic natural disasters break thresholds with little or no warning resulting in long-term modifications in environmental characteristics or functioning of the ecosystem. Of special interest is the work of Steffen et al. (2018) who consider earth biosphere as a whole system and study its possible trajectories under the current planetary crisis. In particular, they explore the risk of self-reinforcing feedback that could eventually push the Earth’s biosphere system to a planetary threshold that, if crossed, could prevent climate stabilization near the Holocene temperature regime (the pre-industrial conditions set out in the Paris Agreement). In the worst case scenario Earth could be driven into the ongoing warming track of a “Hothouse Earth” path, even though human emissions were lowered.

As in other papers reviewed, Sidle et al. (2013), state that “if a system is viewed as resilient, it is generally perceived as remaining within specified bounds, probably close to the optimal operational points” mentioned in Equihua et al. (2014). In there, the authors set again the question of which should be the variables under the “bounded ecosystem” and how to determine the range of values to consider the ecosystem as resilient. More to the point, how much time should be spanned between an ecosystem perturbation for the resilience variable returning to their bound limits? In principle, we consider that this should be in the same order of magnitude that the—natural characteristic time scale of the ecosystem. But once again, the measurement of characteristic time scale for an arbitrary state variable of the ecosystem is an open question. The main problem is that in most cases we will not have a mechanistic model for the variable in question but time-series only. In Abe et al. (2005) the authors use the Wigner function to explore if there is an special time scale under which the system reaches an optimal representation. For multiple time series observed, they contrasted entropy values covering a variety of distinct time domains. For their natural characteristic time, they found that entropy is highly likely to be minimal, implying minimum uncertainty in time-frequency space. Another alternative might be to consider the τ0 time in which the system’s memory tends to be zero, defined by the absolute τ time value for which the C(τ) auto-correlation function crosses the horizontal axis (Fossion et al., 2010). But again, remember returning time is not as important for ecosystem resilience in full dynamic consideration.

Complexity perspective

The Filotas and co-workers reviewed article (Filotas et al., 2014) provides a remarkable introduction to complexity. The authors decompose complexity into eight features an then goes to relate them into a new narrative for forests resulting in an interesting connection with ecosystem resilience and integrity. Generally speaking, a system is complex either by presenting a sufficient number of components with strong enough interaction or if it changes in a velocity comparable to the observer’s time scale, and in most cases both. Forests as a system and forest management, certainly occupy a high position in the complexity gradient.

The authors focus on forests, but clearly what they describe applies to all types of ecosystems. Nevertheless, forests are a good model because they are both widely and intensively managed, and also because they are deeply coupled with human systems. This approach can thus assist forest scientists and managers in conceptualizing forests as integrated socio-ecological systems providing concrete examples of how to manage forests as complex adaptive systems.

There are at least 800 different definitions of what a forest is. Some of them are used simultaneously in the same country for different purposes or scales (Lund, 2006). This is in part because forest types differ widely, depending on factors such as latitude, climate patterns, soil properties, and human interactions. It also depends on who is defining it. An economist could describe a forest in a very distinct manner to a forester or a farmer, in accordance with their specific interests. One of the most widely used definitions is that by FAO (1998), that defines a forest as “a portion of land with area over 0.5 ha, tree canopy cover larger than 10%, which is not primarily subject to agricultural or other specific non-forest uses.” For young forests or regions where tree growth is suppressed by climatic factors, trees should be capable of reaching a height of at least 5 m in situ while meeting the requirement for canopy cover. In general, forest definitions are based on two different perspectives. One, associated with quantitative cover/density variables such as minimum area cover, minimum tree height, or minimum crown size. The other, relates to characteristic spatial features of the territory such as the presence of plantations, agricultural activities or non-forest trees within the forest itself (Kleinn, 1991, 1992, 2001; Lund, 2006; Vidal et al., 2008). The issue is that when natural forests are significantly degraded or superseded by plantations, essential ecosystem services such as CO2 sequestration may be lost; but they may technically continue to be classified as forests under many definitions.

Thus, a fundamental key characteristic of ecosystems is essentially missing in most forest definitions, its complexity. The authors point out that complex systems science provides a transdisciplinary framework to study systems characterized by (1) heterogeneity (2) hierarchy (3) self-organization (4) openness (5) adaptation (6) memory (homeostasis?) (7) non-linearity, and (8) uncertainty. These eight characteristics are shared by complex systems regardless of the nature of their constituents and the article exemplifies them in “forest terms.” The conclusion they reach is that complex systems approach has inspired both theory and applied approaches to improve ecosystem resilience and adaptability is most relevant to our article. While forests are prime examples of complex systems (Perry, Oren & Hart, 1994), forest ecology and management approaches are only starting to incorporate complexity which is seldom invoked in the field.

Heterogeneity or how ecosystem components are distributed in space is important from several points of view. For example, the spatial distribution of resources imposes restrictions on animal foraging. It has widely been proved that the foraging patterns of a variety of animals involve many spatio-temporal scales, as described by Lévy walks (Viswanathan et al., 1999; Ramos-Fernández et al., 2004; Miramontes, Boyer & Bartumeus, 2012). This statistical behavior is present even in human movement patterns (Brown, Liebovitch & Glendon, 2007), and has been linked to evolutionary advantages in terms of search strategies in complex environments (Bartumeus, 2007). Further more, resources distribution patterns induce foraging behaviours linked to seed dispersal. Those patterns feedback into the ecosystem dynamics and influence the distribution of resources in time (Boyer & López-Corona, 2009). In the reviewed article, the authors point out that human influence on ecosystems may reduce ecosystem complexity by altering spatial heterogeneity. For instance, forest cover and resident biota have been homogenized by intensifying and standardizing cultivation methods within and between woodlots. Changing these patterns can significantly influence the capacity of the landscape to maintain materials and energy effectively processed and host the region’s biota; this change in turn decreases its integrity, resulting in a loss of resilience. Even more, in the context of the Anthropocene (Steffen, Crutzen & McNeill, 2007) or Technocene (López-Corona, Ramrez-Carrillo & Magallanes-Guijón, 2019), Human impact in extreme cases may modify forest heterogeneity generating new ecological patterns (niche construction) and interactions, without historical equivalents (Seastedt, Hobbs & Suding, 2008).

Spatial heterogeneity can be altered by invasive species threaten biodiversity through predation (Doherty et al., 2015; Rayner et al., 2007), competition (Harris & Macdonald, 2007), disease transmission (Wyatt et al., 2008), and facilitation of the establishment of further invasive species (Simberloff, 2011)). It has been reported that the decrease and extinction of native species due to invasive predators can generate cascade effects that extend through the whole ecosystem and beyond. (Courchamp, Chapuis & Pascal, 2003). In particular predation effects resulting from human introduced species can be severe (Silva-Rodríguez & Sieving, 2011; Doherty et al., 2016). Both rats (Rattus rattus), cats (Felis catus) and dogs (Canis lupus familiaris) are recognized as the worst threat species following recent studies (Hughes & Macdonald, 2013). In natural areas worldwide, dogs are threatening some 200 species, some of them even included in IUCN threat categories. Likewize, Feral cats and red fox (Vulpes vulpes) predation processes has been documented as a cause of the decline or extinction of two thirds of Australia’s digging mammal species (Fleming et al., 2014; Woinarski, Burbidge & Harrison, 2015). Reduced disturbance to soil in the absence of digging mammals has led to impoverished landscapes where little organic matter incorporates into the soil and rates of seed germination is low (Fleming et al., 2014). The predation of seabirds through introduced Arctic foxes (Alopex lagopus) in the Aleutian archipelago has reduced nutrient input and soil fertility, eventually causing vegetation to shift from grassland to dwarf shrubland.

In a recent work (Dannemann, Boyer & Miramontes, 2018), a deep relation between Lévy walks and ecosystem resilience has been shown. In this work, Danneman and coworkers unveil an essential yet unexplored multi-scale movement property of Lévy walks, how it play an important role in the stability of populations dynamics.Using Lotka–Volterra models, they predict that generally diffusing foragers tend to become extinct in fragile fragmented habitats, while their populations become resilient to degraded circumstances and have maximized abundance when individuals undertake Lévy flights. Their analytical and simulated findings, change the scope of multi-scale foraging from individual to population level, making it of major value to a wide scope of applications in biology of conservation. Their findings indicate that Lévy flights reach a balance between exploration and exploitation, which in turn will benefit the stability and resilience of the population. In that way, modern forest management is becoming more compatible with this complexity feature (heterogeneity) by promoting it through strategic cuts that emulate natural disturbances; leave intact some structures and organisms, Including dead and living trees and intact patches of forests; and promote mixtures of tree species. These methods are similar to the comparatively recent strategy of using biodiversity to boost yield and resilience in natural and managed ecosystems (Filotas et al., 2014). Generalizing these ideas we reckon there is important evidence suggesting that in order to preserve ecosystem integrity and resilience, management systems should consider maintaining minimum levels of ecosystem complexity.

Of course, this poses a new challenge: How to measure complexity? Following Gershenson & Heylighen (2003) one may measure complexity using the Shannon information. In this information theory framework, in order to have new information, the old one has to be transformed. Thus, we can define information emergence E as the rate of information transformation. therefore emergence is identified directly with Shannon’s information H or I. In addition, self-organization (S), a key feature of complex systems, has been correlated with an increase in order (i.e., as a reduction of entropy) (Gershenson & Heylighen, 2003). Thus, if emergence implies an increase in information, which is analogous to entropy; self-organization should be anti-correlated with emergence in such a way that (12) S=1−I=1−E

In this way, following (Gershenson & Heylighen, 2003; Fernández, Maldonado & Gershenson, 2014) complexity can be measured as.

(13) C=4⋅E⋅S

Under the complex systems perspective ecosystems are not systems that can simplistically be managed top-down. We must explicitly consider that the interactions take place in multiple and hierarchical levels. This is a general feature of complex systems, components are organized hierarchically in such a way that elements at different levels interact to form an architecture that characterizes the system. In this way, complexity asserts that a phenomenon occurring at one scale cannot be understood without considering cross-scale interactions. But it also means that environmental policy, management and intervention needs to be rethought in terms of scale. In this respect Taleb is assembling “Principia Politica” (draft version available at: https://www.academia.edu/38433249/Principia_Politica) which include the understanding of policies as scale dependent, and so we should consider that instead of aiming at one monolithic policy for managing ecosystem, we should go on to develop a range of them linked to different levels of application. Such approach will be required to reduce the risk of catastrophic hidden effects.

Understanding the coupling of natural and human sub-systems provide a whole new narrative that challenges management. Ecosystems management is the outcome of collective actions among different agents such as decision makers, scientists, managers, concerned citizens and so on. As complexity (key for ecosystem integrity and resilience) is a dynamic balance between emergence and self-organization (S) (Eqs. (2) and (3)), some (and the correct type of) self-organization is necessary to be fostered, but too much of it is bad. Too much (form the wrong type) of S may sustain unwanted feedbacks with detrimental consequences. For instance, illegal logging in Borneo can be seen as a self-organizing phenomenon supported by interactions among all levels in the stakeholder hierarchy (Putz & Chattaraj, 2013). The mechanism is explained by Filotas and co-workers. The feedback starts with pit sawyers taking out livings and pirate loggers taking advantage of governance failures. This alone could not generate such a great impact unless it couples with unscrupulous timber buyers and corrupt governmental officials laundering the illegal wood. Experience in Mexico suggests that corruption might be the common link in practically all-important ecosystem degradation processes. Finally, they say that savvy international traders are the higher link that provides lucrative outlets for ill-gotten goods. According to the author’s narrative, where illegal logging occurs, wood markets are flooded, wood prices are depressed, and standing trees are undervalued. Under such conditions, community forest managers are not motivated to implement sustainable forest management practices, which often involve short-term investments for only long-term returns. These conditions result in a self-organized feedback that sustains illegal logging.

The characteristic of complex systems of not having a unique description scale is related to one of the most omnipresent system-wide phenomena, the 1/f behavior on frequency space for the fluctuation time series. This so-called pink noise is one of a family of 1/fβ colored or fractal noises defined by the β scaling exponent and considered as criticality fingerprint. It is common to comparing and classifying fluctuation dynamics according to their resemblance to three archetypal noise groups: white (β ∼ 0), pink (β ∼ 1) and Brownian (β ∼ 2) (Bak, Tang & Wiesenfeld, 1988; Landa et al., 2011; Kleinen, Held & Petschel-Held, 2003; Peng et al., 1994). The universality of criticality is still under consideration and is known as the “criticality hypothesis” which states that systems in a dynamic system that shifts between order and disorder reach the greatest level of computing capacities when reaches a balance between robustness and flexibility (see Roli et al., 2018 and references therein). This idea of criticality was recently used in an information theory approach to defining ecosystem health and sustainability (Ramírez-Carrillo et al., 2018). In this article, Ramírez-Carrillo et al. (2018) consider that an ecosystem is healthy if it is in criticality, as a mixture of scale invariance (as power laws in power spectra) and a balance between adaptability and robustness.

These power laws appear in numerous phenomena including earthquake statistics, solar flares, epidemic outbreaks, etc., as summarized by Ramírez-Carrillo et al. (2018), Mandelbrot (1982), Newman (2005) and Sornette (2006). They also are a common theme in biology (Goldberger, 1992; Goldberger, Peng & Lipsitz, 2002b; Gisiger, 2001; West, 2010). Several researchers reported evidence of dynamic criticality in physiological processes such as heart activity and suggested that it could be a main characteristic of a healthy state (Kiyono et al., 2005; Ivanov et al., 1996; Rivera et al., 2016a). Some studies (Goldberger, Peng & Lipsitz, 2002b; Rivera et al., 2016b) found compelling evidence of a relationship between healthy hearts and scale-invariant noise, around 1/f regime, backed by medical evidence.

This complexity approach is clearly complementary to the Ecosystem Integrity narrative, and we should consider that an ecosystem is resilient if, in addition to maintaining its EVLs values inside a “safe” range, it also keeps them within a critical dynamic region (scale invariance and “1/f” fluctuations) (Ramírez-Carrillo et al., 2018).

Beyond resilience, antifragility

Living systems can and must do much more than merely react to the environment’s variability through random mutations followed by selection; they most certainly have built-in characteristics that enable them to discover surrounding variations and cope with adversity, variability and uncertainty. Antifragility is one, maybe the core one of these characteristics (Danchin, Binder & Noria, 2011; Taleb, 2012).

If one considers what does really mean that something is fragile, the key property is that it gets damaged by environmental variability. Now if we ask our nearest colleague at random, about the exact opposite of fragile, most likely we would get concepts such as robustness or resilience. But at close inspection it is clear that none of them are the exact opposite of fragile. Both represent systems that are insensitive to environmental variability or get affected only momentarily, quickly returning to its initial state.

The exact opposite of fragility is defined by Taleb as antifragility, which is a property that enhances the system’s functional capacity to reply to external perturbations (Taleb, 2018). In other words, a system is antifragile if it benefits from environmental variability, works better after being disturbed. Then, antifragility is beyond robustness or resilience. While the robust/resilient systems tolerate stress and remain the same, antifragile structures not only withstand stress but also gain from it, learn or adapt. The immune system provide significant illustration of antifragile systems. When subjected to various germs at a young age, our immune system will improve and gain different capabilities to overcome new illnesses in the future (Pineda, Kim & Gershenson, 2018).

A formal definition of antifragility as convexity in the payoffs space is found in Taleb & Douady (2013) and Taleb (2018). Lets consider a two times continuously differentiable “response” or payoff function f(x). Then the function’s convexity will be defined by the relation ∂2f∂x2≥0 which can be simplified under the right conditions to 12[f(x+Δx)+f(x−Δx)]≥f(x). Then the response function f will exhibit non-linearity to dose, which means that a dose increase will have a much higher impact in relation to this increase. Taleb generalizes this result to a linear combination for which ∑αi=1,0≤αi≤1 in such a way that ∑[αif(xi)]≥f[∑(αixi)]. Again simplifying the argument, under the correct conditions we end up with f(nx) ≥ nf(x). This way, if X is a random variable with support in where the function f is well behaved, and f is convex, we get Jensen’s Inequality.

(14) E(f(x))≥f(E(x))

Without loss of generality, if its continuous distribution with density φ(x) and support in belongs to the location scale family distribution, with φ(x/σ) and σ > 0, then, with Eσ, the mapping representing the expectation under a probability distribution indexed by the scale σ, we have: (15) ∀σ2>σ1,Eσ2[f(x)]≥Eσ1[f(x)]

This way, Taleb defines local antifragility as “a situation in which, over a specific interval, either the expectation increases with the scale of the distribution as in Eq. 3, or the dose-response is convex over the same interval.”.

Although antifragility framework was developed by Taleb in the context of financial risk analysis, duo to its universal mathematical formalism it has track attention and has been applied far away its original scope. There are applications of the antifraglity concept from molecular biology to urban planning (see Pineda, Kim & Gershenson, 2018 and references inside). In their work, Pineda and co-workers (Pineda, Kim & Gershenson, 2018) proposed a straightforward implementation of antifragility by defining as payoff function the complexity of the system, which makes a lot of sense in the context of our review because complexity is highly related with critically and hence with the trade-off balance between robustness and adaptability.

The authors defined fragility as (16) ∮=−ΔC|Δx|

where ΔC is the change in system complexity due to a perturbation of degree |Δx|. As complexity can always be normalized to, then positive values of ∮ define fragile systems; when ∮ is zero the system is robust/resilient; and for negative values of ∮ the system is Antifragile.

Then, Pineda, Kim & Gershenson (2018) apply it to random boolean networks (RBNs) of a model of genetic regulatory works. They found that ordered RBNs are the most antifragile and demonstrated that, as expected, seven biological well studied networks such as CD4+ T cell differentiation and plasticity or Arabidopsis thaliana cell-cycle, are antifragile.

We know, from central limit theory, that normal distributions can only emerge from (simple) systems without interactions (probabilistic independence). When we take into account interactions (no probabilistic independence) then the corresponding probability distribution will have fat-tails. In that sense complexity is related with fat-tails and fat-tails with fragility/antifragility (Taleb, 2012). In Taleb’s narrative, normal distribution in the response function characterize robust systems; whereas left fat-tailed are fragile, and right fat-tailed are antifragile systems. Most interestingly, Fossion, Rivera & Estanol (2018) have related homeostasis (physiological resilience?) to pairs of physiological variables, one to be controlled (the one that remains in homeostasis) and another one that controls the former. The main idea is that in order to have a homeostatic physiological variable (normal), the body must use other variables (right fat-tailed) to absorb a random injection of matter, energy, information or any combination of them from the environment. In Fig. 4 of their article, Fossion, Rivera & Estanol (2018) present results for variability analysis of heart rate HR and blood pressure BP for (a) healthy control(s), (b) recently diagnosed diabetic patient(s) and (c) long-standing diabetic patient(s). They show that for healthy patients BP is normal and HR is right fat-tailed. In the case of recently diagnosed diabetic patients, BP start to lose normality and develop a left tail and HR tends to normality. Eventually, long-standing diabetic patients, BP has a clearly left fat-tailed behavior and HR has become normal. This is very compelling evidence of the role of antifragility in human health.

Figure 4 Basic characteristics of systems in terms of antifragility, which is the property of a system to respond in a convex way to perturbations or variability.

(A–C) are examples of fragile, robust/resilient and antifragile systems respectively; (D–F) are examples of profile responses to perturbations; (J–L) are examples of typical probability distributions; and (M–O) are the characteristic values obtained with the metric based on complexity change.

In a general manner, Taleb (2012) suggests the so-called barbell (or bimodal) approach as an archetypal strategy for achieving antifragility. The first step towards antifragility is to reduce downsides instead of increasing upsides. In other words, by reducing exposure to adverse low probability but elevated adverse payoff occurrences (i.e., “black swans ” events) and allowing natural antifragility to function on its own. We follow Taleb with a vulgar finance instance, where the idea is easiest to explain, although most of them are misunderstood. The barbel approach in finances comprises of placing 90% of your resources in safe instruments (provided that you are protected against inflation) or what is referred to as the “value repository number,” and 10% in very risky, maximal risky bonds, exposing yourself to unpredicted massive gains in a convex way. In this form, one ends up with some sort of bimodal optimization taking advantage at the same time of the robustness of safe inversion and on the other hand the adaptability of high risky ones. Anyone who has a 100% stake in so-called “medium” securities (unimodal optimization) is at danger of “complete risk ruin.” This Barbel Strategy addresses the issue of incomputability and fragility in the assessment of the hazards of unusual occurrences.

As in the barbell strategy, a basic mechanism to achieve antifragility, is a thorough strategy to risk management under fat-tailed distributions and as those are widely present in nature, then it should be very ubiquitous in natural systems. We identify this barbell risk strategy as the “good balance” property in network topology by means of the relation of strong and weak interactions or between ascendancy and overhead; and balance robustness and adaptability, identified as fingerprint of critically (scale invariant and 1/f type of noise), in the dynamic of system’s fluctuations. For us is clear that all these three “good balances” are related and as we have said early in the text, it is very plausible that all these three kinds of balance could be particular cases of a more general evolutionary strategy of living systems: the antifragility.

Discussion

From the analysis of the literature, we found that the citation network from reviewed articles is not percolated, what we interpret as a lack of unification in the research field and an opportunity for interdisciplinary work (see Supplemental materials).

We found (see Fig. 2) three main narratives (a) ecosystem properties that enable them to be more resilient; (b) ecosystem response to perturbations; and (c) complexity. From this and complementary literature consulted we have identified 11 possible indicators for ecosystem resilience (see Table 3 and a glossary in Table 4). In particular we showed how to apply Fisher information in a study case which we consider a very promising proxy of resilience, since it has a solid formal framework, it is easy to implement and it can be applied to any kind of system.

Table 3 Resilience measure found in the literature review and complementary papers.

Key	Indicator	Measure/proxy	Requires	Resilience	Narrative	
FI	Fisher information	Stability	Time series	More stable ecosystem are more resilient and according to Cabezas et al. (2005) for a system to be resilient, after a disturbance the FI values prior to it must be recovered	Perturbations	
Div	Diversity	Optional/use of resource space.	Presence field data	In general to greater diversity, greater resilience. But there are exceptions related to changes of composition and use of resources	Properties	
Co	Network conectance	Stability	Knowing the networks and being able to quantify the intensity of the connections, Gustavson proposes ways to deal with the lack of information about it	Increase in the number of connections dissipates the effect of variation in distribution of species and enhances stability species	Properties	
Omn	Presence of omnivore species	Communication between different scales	Presence of omnivore species	Presence of omnivore species enhance stability and resilience	Properties	
NC	Network criticality	Balance between robustness (strong Interactions) and adaptability (Weak Interactions).	Knowing the networks and being able to quantify the intensity of the connections, Gustavson proposes ways to deal with the lack of information about it	Observations show that ecosystems are more resilient when there is a good balance between the number of strong and weak connections	Properties	
L-VC	Lotka–Volterra Coefficients	Given a community matrix, if all the real parts of its
eigenvalues are negative the ecosystem is stable	Community matrix	More stable ecosystem are more resilient	Complexity	
As	Ascendency	Mean mutual information	Given a network of interactions (i.e., trophic network) it measures how well, on average, the network articulates a flow event between any two nodes	Capture in a single index the ability of an ecosystem to prevail against disturbance by virtue of its combined organization and size	Properties	
Lévy	Lévy flights	Scaling coefficient of foraging patterns for key species such as puma or jaguar	It is a proxy of resources spatial complexity	It has been shown that Lévy flights foraging patterns are related and enhance ecosystems resilience	Complexity	
Frac	Fractality	Spatial complexity	High resolution satellite images	More complex ecosystems should be more resilient.	Complexity	
AF	Antifragility	Change in the complexity of a biotic (i.e., trophic) network, in the face of disturbances	Network of interactions, can be a Boolean network of co-occurrences of a key species such as puma or jaguar with its prey for example	Resilience would be an intermediate state between fragility and antifragility	Perturbations	
H	Homeostasis	System homeostasis	Time series	Equivalent of resilience	Complexity	

Table 4 Glossary for uncommon terms used in this article.

Term	Definition	
Antifragility	Antifragility is a property that enhances the system’s functional capacity to response to external perturbations (Taleb, 2018). In other words, a system is Antifragile if it benefits from environmental variability, works better after being disturbed	
Ascendency	It is a measure of the magnitude of the information flow through an ecosystem’s network framework	
Complexity	A system is complex either it presents a sufficient number of components with strong enough interaction or it changes in a velocity comparable to the observer’s time scale, and in most cases both. Forests as a system and forest management, certainly occupy a high position in the complexity gradient (Filotas et al., 2014). It is measure as the product of emergence and self-organization	
Criticality	Criticality is a regime in which the system is in dynamic scale invariance (power law in frequency space) and in an “optimum” balance between robustness and adaptability (scale coefficient around −1)	
Emergence	We can define information emergence E as the rate of information transformation. Can be measure as Shannon information	
Fisher information	Fisher information may be understood as the quality of a measurement-inference process. It is related to tangential velocity and acceleration in phase space, hence with stability	
Homeostasis	Fossion and co-workers (Fossion, Rivera & Estanol, 2018) have related homeostasis (physiological resilience?) to pairs of physiological variables, one to be controlled (the one that remains in homeostasis) and another one that controls the former. The main idea is that in order to have a homeostatic physiological variable (normal), the body must use other variables (right fat-tailed) to absorb a random injection of matter, energy, information or any combination of them from the environment	
Integrity	Is a measure of the state of the ecosystems in terms of its structure, composition and function	
Lévy-flight	Fat-tailed foraging pattern characterized by local space exploration (normal distributed) with some large “flights” for non-local exploration	
Persistence	Persistence is the time for a variable to remain in the same state before changing to a different one (Pimm & Pimm, 1991). Persistence is a measure of a system’s capacity to preserve itself over time (Loreau et al., 2002)	
Robustness	Robustness relates to the durability of the stability of the environment. Robustness is then a measure of the amount of disturbance an ecosystem can endure before it changes to a different state (Loreau et al., 2002). The more robust the food web is, the more stable it is	
Self-organization	Is the complement of emergence (1 − E) and represent the capacity of the system for increase its organization	

Nevertheless a new way to reinterpret resilience emerged from this critical literature review: Antifragility. This novel framework developed by Taleb (2012), Taleb & Douady (2013) and Taleb (2018) is based on fat-tailed, non-linear responses of the system to variability (see Fig. 4). In a simple way, if a system has a concave (non-linear) payoff function dependent of certain variable, then the system is fragile to it. On the contrary, if the payoff in convex then it is Antifragile and if the system is essentially insensible to variability, then is robust/resilient. In Taleb’s work, antifragility is associated with bimodal risk strategy called “The Barbell” which we believe manifest itself as “a good balance” between: (i) strong and weak interactions in network topology; (ii) adaptability and robustness (Criticality); and (iii) ascendancy and overhead.

Real ecosystem antifragility may me measure using the complexity based metric presented inhere using time series permutation entropy as in (Ramírez-Carrillo et al., 2018) and then the complexity formulae from Gershenson & Fernández (2012). In addition to this direct metric, as pointed out by one of the manuscript reviewer, we consider Right-skewness in ecosystem state variables distributions may be one indicator of Ecosystem antifragility. The intuition for this came from recent work by Fossion, Rivera & Estanol (2018) in which they propose that in human systems (but we think is a general result), vital state variables for the organism as blood pressure need to be maintain into a very sharp range of values following a Gaussian distribution. This is no new since correspond to traditional ideas of homeostasis, the novel thing is that they identifies that in order to achieve this, organism need absorb energy, matter and information fluctuations from the environment. This absorbing or controlling process is carried out by heart activity variability. In this way, youth and health are characterized by Guassianity in homeostatic physiological variables and Right-skewness for absorbing (controlling) ones. In their work, the authors show how under chronic disease such as diabetes that statistical behavior flips: blood pressure (homeostatic variable) is Left-skewness distributed and Heart Rate (absorbing variable) become Gaussian (Fossion, Rivera & Estanol, 2018). Some problems could be to identified pairs (or network) of homeostatic-absorbing variables, and if skewness may be affected by other processes other than fragility.

Although our main interest was in the Informational Theory approach, along the peer review process we were suggested to grow the scope of our search criteria to capture more works in which Integrity and Resilience interact without Information Theory in between. We then conducted a new search using "Resilience AND Integrity AND Ecosystem" but asking for these keywords to be in the title in order to ensure the papers were focused on them and not treat them only in a marginal way. We found three new papers Müller, Burkhard & Kroll, 2009; Bratanova-Doncheva et al., 2014; Chipev et al. (2013) which fall into our previously identified narratives.

The first one makes an interesting discussion about the importance of studying how the ecosystem responds to perturbations for understanding ecosystem dynamics. The authors review, for example, the historical evolution of the subject based on the work on the 30’s where complexity (in the way it was understood at the time, most likely as self-organization) was identified as a core precursor of stability. By the 50–60’s the authors highlight the importance of equilibrium through three major components: stability, resistance, and robustness (see Fig. 1 in Grafton et al. (2019) for a good explanation of these terms). They say that in 60–70’s literature started to mention vulnerability, fragility, path dependence, and local instability, arriving to disequilibrium in the 80–90’s which gave place to the Holling perspective of resilience, updated later in 2012 by incorporating the concept of panarchy, defined as “a temporal structure in which the system is interlinked in continual adaptive cycles of growth, accumulation, restructuring, and renewal” Gunderson, Allen & Holling (2012). The authors put forward the idea that resilience and ecosystem dynamics change in different stages of ecosystem development, starting in a dynamics condition governed by components optimization (orientors) throughout undisturbed successions. They argue that in this context, integrity indicates orientor development but warns that integrity evaluations do not fully characterize ecosystem dynamics because it does not inform about how the ecosystem responds to perturbations. They further argue, for example, that in general it can be concluded that the ability to adapt after changes of the constraints may be decreased when a high degree of maturity is attained where mutual inter-dependency is maximum. This is a feature observed also in human induced ecosystems for which older individuals depart from criticality by an emergence loss so they are too self-organized (Goldberger et al., 2002a), a characteristic Perrow Perrow (2011) identified as necessary for a system to experiment normal accidents or be exposed to inevitable systemic accidents.

The other two papers relate integrity and resilience to ecosystem services so, they fall within our ecosystem properties narrative. Although ecosystem services could be considered a different narrative, it is not really defined for ecosystems but for socio-ecosystems. Thus, it properly should be a narrative framed within a socio-ecosystem resilience context, which we will briefly discuss in the next section.

Conclusions

In the long term, considering the coupling of ecosystem with human systems (i.e. via climate change) we consider that antifragility is a more desirable feature than resilience. Thinking in socio-ecosystems, we can see that they usually not only keep on living, but they do flourish and evolve, even in the presence of great stressors such as climate crisis or land change. In fact, in a recent work (De Bruijn, Größler & Videira, 2019), It has been shown that the outcome of using antifragility as a design criterion is that the scheme being studied demonstrates a more favorable behavior than a "simply" robust model in a setting that is susceptible to black swans (unpredictable, very low frequency of ocurrancebut vey high impact events). Then, for socioecosystem governance, planning or in general, any decision making perspective, antifragility might be a valuable and more desirable goal to achieve than a resilience aspiration (Blečić & Cecchini, 2018). On the other hand, naive interventionism of natural systems, such as randomness or stressor suppression may lead to a fragilization process as widely discuss by Taleb (2012). Living systems need the correct amount of randomness to flourish as showed by Fernandez-Oto, Tzuk & Meron (2019) with dryland ecosystems which in the presence of nonlinear front instabilities may enter into a self-recovery process from desertification. This makes us question: What may be the relation of Criticality, Antifragility and Fisher information? and How antifragility concept may enter into for example Cabezas et al. (2005) or Ramírez-Carrillo et al. (2018) sustainability frame work?

Supplemental Information

Supplemental Information 1 Suplemental materials.

Click here for additional data file.

Additional Information and Declarations

Competing Interests

Author Contributions

Data Availability

The authors declare that they have no competing interests.

Miguel Equihua Zamora conceived and designed the experiments, analyzed the data, prepared figures and/or tables, authored or reviewed drafts of the paper, and approved the final draft.

Mariana Espinosa Aldama conceived and designed the experiments, performed the experiments, analyzed the data, prepared figures and/or tables, authored or reviewed drafts of the paper, and approved the final draft.

Carlos Gershenson conceived and designed the experiments, authored or reviewed drafts of the paper, and approved the final draft.

Oliver López-Corona conceived and designed the experiments, performed the experiments, analyzed the data, prepared figures and/or tables, authored or reviewed drafts of the paper, and approved the final draft.

Mariana Munguía conceived and designed the experiments, prepared figures and/or tables, authored or reviewed drafts of the paper, and approved the final draft.

Octavio Pérez-Maqueo conceived and designed the experiments, prepared figures and/or tables, authored or reviewed drafts of the paper, and approved the final draft.

Elvia Ramírez-Carrillo conceived and designed the experiments, performed the experiments, analyzed the data, prepared figures and/or tables, authored or reviewed drafts of the paper, and approved the final draft.

The following information was supplied regarding data availability:

This article is a literature review and analyzed the literature.

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
