# Peer review of "Ecosystem antifragility: beyond integrity and resilience"

_PeerJ, doi:10.7717/peerj.8533_

## Round 0.1 · original submission · Major Revisions

I tend to agree with Reviewer 2 who has provided very detailed comments. The major issues are a central organizing principle, general text organization, and some limitations on the analysis. In many places language can be clarified as noted. I also agree that while resilience and development are well-explored topics across disciplines some of the ideas here, especially anti-fragility, are important additions to the discussion. I also agree that the interpretation of resilience is somewhat flawed by the focus on integrity. A key component of resilience is flexibility but full integrity does not need to be maintained for a system to be resilient. Please do follow the suggestions of the second reviewer closely and that should be sufficient to address the one concern noted by the first reviewer. The paper is a potentially interesting contribution to the literature and I welcome a revision if these problems can be addressed.

Reviewer 1 ·

Basic reporting

no comment

Experimental design

no comment

Validity of the findings

no comment

Additional comments

Nice review paper for interdisciplinary scientist thinking about stability and resilience of ecosystem. After presenting many definitions of resilience measure from the literature, it attempts to ask readers to think resilience in terms of anti-fragility, where the fitness function is an increasing function to environmental perturbations. In other words, as the authors put it, anti-fragile system flourishes and evolve in the presence of environmental stress/perturbation like climate change.

I read the paper as if anti-fragility might be more desirable measure than other existing definition of resilience. Thus, one possible improvement would be to also consider cases where anti-fragile system might be less desirable.

Reviewer 2 ·

Basic reporting

Please see general comments to the author for details.

I believe the language of the article needs work, particularly in paragraph and section organization. I also ask the authors to proof-read the work again for missing references and words. I believe more work on the Introduction is needed to better motivate the study.

The results are novel, interesting, and fitting for the journal but need much work to increase clarity.

Experimental design

Please see general comments to the author for specific comments and suggestion on organization and methods.

Validity of the findings

The results of literature review need to be organized and presented more clearly in order for the argument of the paper to be clearly supported and understood. Please see general comments to the author for details.

Additional comments

This paper reviews the insights of information theory approaches to ecosystem resilience and suggests the potential presence of “antifragile” ecosystems that benefit from fluctuations, i.e. ‘good years help more than bad years hurt’. I find the latter insight particularly interesting and valuable if the authors can better elaborate it using a simple example and illustrate its potential (i.e., concave-up relations) based on the current literature. Following this, my three major comments are:


1 - Unclear motivation, methods choice, and results from the literature review. I struggled with this section partly due to my limited knowledge of information theory, but especially due to the unclear question and organization here.

Introduction/motivation
1.1 - Whilst “ecosystem integrity” is clearly defined, it is not clear how/if this aspect is missing from longstanding theoretical work on ecological resilience. Ecological theory typically defines “ecological resilience” as the the magnitude of perturbations that ecosystems can withstand and return to the target state, as opposed to shifting into an alternative stable state. If a shift from one ecosystem steady state to another is different from losing “ecosystem integrity”, please clarify and emphasize this difference. If it is not different, consider focusing on “ecological resilience” rather than (engineering) “resilience” and “integrity” throughout.

1.2 - the overall focus on information theory needs to be motivated in the introduction - what does an information theory-oriented approach provide that conventional metrics of resilience do not? As this question implies, a comparison with conventional resilience metrics in the literature review could be very helpful.

1.3 - After motivating the information theory approach to resilience generally, the Introduction needs to motivate the specific question of the review. Is it to introduce the readers to potential uses of information-based metrics of resilience? Or the findings of resilience studies that use information theory? Both are valuable but a specific focus is critical for the paper to have a clear goal.

Methods
1.4 - I do not understand why the authors limit the meta-analysis of information theory-based resilience analysis only to papers that also mention “ecosystem integrity”. This seems to needlessly and severely constrain sample size (7 papers!) based on an arbitrary metric of whether the “integrity” language was used. To be clear, I think “integrity” is well-defined in this paper, but in my experience it is not a universally used term in the way “resilience” and “information theory” are.
I suggest (a) use a larger sample size in the literature review by omitting “ecosystem integrity” from the query OR (b) omitting the literature review altogether and just focus on the focal papers you chose.

1.5 - what is the point of figures 4-8? These results are not mentioned anywhere in the manuscript. It may be better to omit or move these figures to an appendix; otherwise, they distract the reader from the main message.

Results
1.6 - I struggled to understand the main findings of this section because many paragraphs simply summarize papers without clearly connecting them to the main narratives. It seems the main results are “We found (see Figure 9) three main narratives (a) Ecosystem properties that enable them to be more resilient; (b) Ecosystem response to perturbations; and (c) Complexity. From this and complementary literature consulted we have identified 11 possible indicators for ecosystem resilience (See Table 5)”. If so, please:

1.6.1: make these 3 narratives the titles of sub-sections. Start each sub-section with a paragraph reviewing the connection between the narrative and ecosystem resilience/integrity or ecological resilience.

1.6.2: as you reference findings from different papers, be careful to convert them to the standard terminology you use in this paper (see major comment #3).

1.6.3: Throughout, rework paragraphs to have clear topic sentences. That is, the first sentence in each paragraph should clearly main point or theme of the paragraph. Clear writing is especially important when communicating unfamiliar (for ecologists) concepts like information theory or frequently misinterpreted concepts such as resilience. For details, please see “Writing Science: How to Write Papers That Get Cited and Proposals That Get Funded” by Joshua Schimel.

1.6.4: In Table 5 (which is a great summary!), please add a column denoting which narrative/feature each metric relates to.





2 - Better clarify the potential for antifragility and its relation to complexity.

2.1 It is unclear how often antifragility might happen in natural systems, especially at the ecosystem scale. The underlying driver of antifragility is a concave-up relation between “environment” and ecosystem condition. However, it is unclear when, if ever, such a relation could arise and have measurable impacts at the scale of entire ecosystems. In population dynamics, for instance, the most common finding is a concave-down relation - that bad years hurt more than good years help. I would like the authors to:

2.1.1 - add a table that includes examples of concave-up ecological relations AND their observed scale - ie, whether the “response” is measured at the individual, population, or ecosystem level. Some ecological examples that might fit here are:
Storage effects (see Chesson, P. (2000). Mechanisms of maintenance of species diversity. Annual review of Ecology and Systematics, 31(1), 343-366.)
Disturbances reducing meta-population or -community synchrony (Fox, J. W., Vasseur, D., Cotroneo, M., Guan, L., & Simon, F. (2017). Population extinctions can increase metapopulation persistence. Nature ecology & evolution, 1(9), 1271.)

2.1.2 - add a Discussion paragraph elaborating how antifragility might be detected in natural systems. Right-skewness in ecosystem state distributions seems to be one indicator, but perhaps you might discuss potential reasons of how processes unrelated to fragility might also affect state skewness, for instance transient dynamics. Alternatively, if there are too many variables confounding skewness, perhaps the answer is that quantifying antifragility requires a validated, mechanistic model of ecological dynamics.


2.2 - Provide a concrete example of “complexity” and how it translates to fragility using a simple ecological model (e.g., of population size). A good model to use may become apparent after assembling the table mentioned above (2.1.1). One simple example might be the relation between physiological performance (e.g., per capita fecundity F) and temperature, which is concave-up at lower temperatures and concave-down at higher temperatures. The population dynamics would then be dn/dt = F(Temperature)*n*(1-n).




3 - Clarify definitions of terms as used in the paper.

3.1 - Please include a table listing all terms found in the paper and their definitions, such as “integrity”, “resilience”, “complexity”, “sustainability”, etc.. In this table you may omit the metrics that you define in Table 5.

3.1 - “resilience” is an oft-confused term in ecology. Throughout, I suggest you use either “engineering resilience” (return time) or “ecological resilience” (see point 1.1). Regardless, please be sure to define both integrity and resilience (or ecological resilience, if you use that instead) from the very beginning (Paragraph 1 of Intro).



Smaller comments:
* How is Figure 9 made?
* Line 65-66: here, please clarify that “persistence time” is only a relevant metric if the current ecosystem state is not globally stable - i.e., that another attractor exists. Much of the current confusion over ecological resilience lies in studies implicitly assuming that in a given system another attractor exists without explicitly defining what it is. Also, here please point out the close relation of “persistence” with “ecological resilience”.
* Lines 63-64 “and assumes that there is only one balance or a stable state”: I do not think “engineering resilience” assumes a single steady state; it just measures the return speed to a given state. So multiple attractors would each have their own engineering resilience.
* Lines 67-69: Currently many ecologists refer to “resistance” as the magnitude of disturbance that a system can withstand without a change in phase (e.g., intensity of drought that causes tree mortality and decline in abundance). Please either change this accordingly or provide a more recent reference for the definition you use.
* Line 71 “Robustness relates to the durability of the stability of the environment”: I don’t understand this sentence.
* Lines 169-171 “to enhance ecosystem’s long-term sustainability, a particularly densely connected network structure is advantageous”: I am not sure what you mean by sustainability in this MS and how it differs from other terms you use. You may need to caveat this statement because in general high connectance (1) can reduce ecosystem stability (“engineering resilience”, see May, R. M. 1972. Will a large complex system be stable? Nature 238:5364) and (2) can allow phase transitions to spread more readily through a system.
* Line 291 “...or it exhibits changes in the configuration space comparable to the observer’s time scale”: this is unclear - please reword more simply.
* Throughout, check for word misspelling and missing references.
* Throughout, please check for mistakes in grammar and word choice. For instance, on lines 290-291 “a system is complex either it presents” I believe should read “a system is complex if either it presents”.

---

## Round 0.2 · Major Revisions

The revised paper has made a stronger argument for an information theory approach in the introductory section and has clarified other key issues regarding how terms are used. The primary concern remains: the literature review is extremely limited. I accept the argument made that integrity and resilience are different but related concepts. I think this distinction is important and appreciate how they are each discussed in the paper. However, as reviewer 2 pointed out, integrity is not universally defined and used. Thus, the inflexibility of the search terms may well be overlooking sources--especially those that might discuss integrity but use different language. Plus, resilience has been of greater interest because it describes a system withstanding stress rather than a stable state (how integrity is often perceived and overlooked as less interesting). A more flexible search that will return more results DOES require more work on the part of the authors to sift through the many to distill the few that hit the target. The result might identification of the same nine papers given the historical focus on resilience over integrity, but I am not convinced yet this is the case. Thus, I am left concerned that the scope of the paper excludes viable candidates for the literature review that might affect the discussion and conclusions. Further, I believe a discussion of papers that include both integrity and resilience would add to the general literature review even if they do not include information theory simply because information theory approaches could be applied to those papers as well. As such, I have returned this for major revisions to address specifically the search parameters. There are other minor revisions that could be made but I will exclude those from this decision as I realize you may wish to stand firm in your resolve to keep your search limited and submit the paper elsewhere.

---

## Round 0.3 · accepted · Accept

The results of the new search are interesting in that very few additional results were returned. Thanks for taking the time to address that issue. The revisions are acceptable.